# BOLSTERING STOCHASTIC GRADIENT DESCENT WITH MODEL BUILDING

## ABSTRACT

Stochastic gradient descent method and its variants constitute the core optimization algorithms that achieve good convergence rates for solving machine learning problems. These rates are obtained especially when these algorithms are fine-tuned for the application at hand. Although this tuning process can require large computational costs, recent work has shown that these costs can be reduced by line search methods that iteratively adjust the stepsize. We propose an alternative approach to stochastic line search by using a new algorithm based on forward step model building. This model building step incorporates a second-order information that allows adjusting not only the stepsize but also the search direction. Noting that deep learning model parameters come in groups (layers of tensors), our method builds its model and calculates a new step for each parameter group. This novel diagonalization approach makes the selected step lengths adaptive. We provide convergence rate analysis, and experimentally show that the proposed algorithm achieves faster convergence and better generalization in most problems. Moreover, our experiments show that the proposed method is quite robust as it converges for a wide range of initial stepsizes.

## 1 INTRODUCTION

Stochastic gradient descent (SGD) is a popular optimization algorithm for machine learning problems which can achieve very fast convergence rates when its stepsize and its scheduling are tuned well for the specific application at hand. This tuning procedure can take up to thousands of CPU/GPU days resulting in big energy costs (Asi & Duchi, 2019).

A number of researchers have studied adaptive strategies for improving the direction and the stepsize choices of the stochastic gradient descent algorithm. Adaptive sample size selection ideas (Byrd et al., 2012; Balles et al., 2016; Bollapragada et al., 2018) improve the direction by reducing its variance around the negative gradient of the empirical loss function, while stochastic quasi-Newton algorithms (Byrd et al., 2016; Wang et al., 2017) provide adaptive preconditioning. Recently, several stochastic line search approaches have been proposed. Not surprisingly, some of these work cover sample size selection as a component of the proposed line search algorithms (Balles et al., 2016; Paquette & Scheinberg, 2020).

The Stochastic Model Building (SMB) algorithm proposed in this paper is not designed as a stochastic quasi-Newton algorithm in the sense explained by Bottou et al. (2018). However, it still produces a scaling matrix in the process of generating trial points, and its overall step at each outer iteration can be written in the form of matrix-vector multiplication. Unlike the algorithms proposed by Mokhtari & Ribeiro (2014) and Schraudolph et al. (2007), we have no accumulation of curvature pairs throughout several iterations. Since there is no memory carried from earlier iterations, the scaling matrices in individual past iterations are based only on the data samples employed in those iterations. In other words, the scaling matrix and the incumbent random gradient vector are dependent.

Vaswani et al. (2019) apply a deterministic globalization procedure on mini-batch loss functions. That is, the same sample is used in all function and gradient evaluations needed to apply the line search procedure at a given iteration. However, unlike our case, they employ a standard line search procedure that does not alter the search direction. They establish convergence guarantees for the empirical loss function under the *interpolation* assumption, which requires each component loss function to have zero gradient at a minimizer of the empirical loss. Mutschler & Zell (2020) assume that the

optimal learning rate along the negative batch gradient is a good estimator for the optimal learning rate with respect to the empirical loss along the same direction. They test validity of this assumption empirically on deep neural networks (DNNs). Rather than making such strong assumptions, we stick to the general theory for stochastic quasi-Newton methods.

Other work follow a different approach to translate deterministic line search procedures into a stochastic setting, and they do not employ fixed samples. In Mahsereci & Hennig (2017), a probabilistic model along the search direction is constructed via techniques from Bayesian optimization. Learning rates are chosen to maximize the expected improvement with respect to this model and the probability of satisfying Wolfe conditions. Paquette & Scheinberg (2020) suggest an algorithm closer to the deterministic counterpart where the convergence is based on the requirement that the stochastic function and gradient evaluations approximate their true values with a high enough probability.

With our current work, we make the following contributions. We use a model building strategy for adjusting the stepsize and the direction of a stochastic gradient vector. This approach also permits us to work on subsets of parameters. This feature makes our model steps not only adaptive, but also suitable to incorporate into the existing implementations of deep learning networks. Our method changes the direction of the step as well as its size which separates our approach from the backtracking line search algorithms. It also incorporates the most recent curvature information from the current point. This is in contrast with the stochastic quasi-Newton methods which use the information from the previous steps. Capitalizing our discussion on the independence of the sample batches, we also give a convergence analysis for SMB. Finally, we illustrate the computational performance of our method with a set of numerical experiments and compare the results against those obtained with other well-known methods.

## 2 STOCHASTIC MODEL BUILDING

We introduce a new stochastic unconstrained optimization algorithm in order to approximately solve problems of the form

$$\min_{x \in \Re^n} \quad f(x) = \mathbb{E}[F(x, \xi)], \tag{1}$$

where $F : \mathbb{R}^n \times \mathbb{R}^d \to \mathbb{R}$ is continuously differentiable and possibly nonconvex, $\xi \in \mathbb{R}^d$ denotes a random variable, and $\mathbb{E}[.]$ denotes the expectation taken with respect to $\xi$. We assume the existence of a stochastic first-order oracle which outputs a stochastic gradient $g(x, \xi)$ of $f$ for a given $x$. A common approach to tackle (1) is to solve the empirical risk problem

$$\min_{x \in \Re^n} \quad f(x) = \frac{1}{N} \sum_{i=1}^{N} f_i(x), \tag{2}$$

where $f_i : \mathbb{R}^n \to \mathbb{R}$ is the loss function corresponding to the $i$th data sample, and $N$ denotes the data sample size which can be very large in modern applications.

As an alternative approach to line search for SGD, we propose a stochastic model building strategy inspired by the work of Öztoprak & Birbil (2018). Unlike core SGD methods, our approach aims at including a curvature information that adjusts not only the stepsize but also the search direction. Öztoprak & Birbil (2018) consider only the deterministic setting and they apply the model building strategy repetitively until a sufficient decent is achieved. In our stochastic setting, however, we have observed experimentally that multiple model steps does not benefit much to the performance, and its cost to the run time can be extremely high in deep learning problems. Therefore, if the sufficient decent is not achieved by the stochastic gradient step, then we construct only one model to adjust the size and the direction of the step.

Conventional stochastic quasi-Newton methods adjust the gradient direction by a scaling matrix that is constructed by the information from the previous steps. Our model building approach, however, uses the most recent curvature information around the latest iteration. In the popular deep learning model implementations, model parameters come in groups and updates are applied to each parameter group separately. Therefore, we also propose to build a model for each parameter group separately making the step lengths adaptive.

The proposed iterative algorithm SMB works as follows: At step $k$, given the iterate $x_k$, we calculate the stochastic function value $f_k = f(x_k, \xi_k)$ and the mini-batch stochastic gradient

$g_k = \frac{1}{m_k} \sum_{i=1}^{m_k} g(x_k, \xi_{k,i})$ at $x_k$, where $m_k$ is the batch size and $\xi_k = (\xi_{k,1}, \ldots, \xi_{k,m_k})$ is the realization of the random vector $\xi$. Then, we apply the SGD update to calculate the trial step $s_k^t = -\alpha_k g_k$, where $\{\alpha_k\}_k$ is a sequence of learning rates. With this trial step, we also calculate the function and gradient values $f_k^t = f(x_k^t, \xi_k)$ and $g_k^t = g(x_k^t, \xi_k)$ at $x_k^t = x_k + s_k^t$. Then, we check the stochastic Armijo condition

$$f_k^t \leq f_k - c\,\alpha_k \|g_k\|^2, \tag{3}$$

where $c > 0$ is a hyper-parameter. If the condition is satisfied and we achieve *sufficient decrease*, then we set $x_{k+1} = x_k^t$ as the next step. If the Armijo condition is not satisfied, then we build a quadratic model using the linear models at the points $x_{k,p}$ and $x_{k,p}^t$ for each parameter group $p$ and find the step $s_{k,p}$ to reach its minimum point. Here, $x_{k,p}$ and $x_{k,p}^t$ denote respectively the coordinates of $x_k$ and $x_k^t$ that corresponds to the parameter group $p$. We calculate the next iterate $x_{k+1} = x_k + s_k$, where $s_k = (s_{k,p_1}, \ldots, s_{k,p_n})$ and $n$ is the number of parameter groups, and proceed to the next step with $x_{k+1}$. This model step, if needed, requires extra mini-batch function and gradient evaluations (forward and backward pass in deep neural networks).

For each parameter group $p$, the quadratic model is built by combining the linear models at $x_{k,p}$ and $x_{k,p}^t$, given by

$$l_{k,p}^0(s) := f_{k,p} + g_{k,p}^\top s \quad \text{and} \quad l_{k,p}^t(s - s_{k,p}^t) := f_{k,p}^t + (g_{k,p}^t)^\top (s - s_{k,p}^t),$$

respectively. Then, the quadratic model becomes

$$m_{k,p}^t(s) := \alpha_{k,p}^0(s) l_{k,p}^0(s) + \alpha_{k,p}^t(s) l_{k,p}^t(s - s_{k,p}^t),$$

where

$$\alpha_{k,p}^0(s) = \frac{(s - s_{k,p}^t)^\top (-s_{k,p}^t)}{(-s_{k,p}^t)^\top (-s_{k,p}^t)} \quad \text{and} \quad \alpha_{k,p}^t(s) = \frac{s^\top s_{k,p}^t}{(s_{k,p}^t)^\top s_{k,p}^t}.$$

The constraint

$$\|s\|^2 + \|s - s_{k,p}^t\|^2 \leq \|s_{k,p}^t\|^2,$$

is also imposed so that the minimum is attained in the region bounded by $x_{k,p}$ and $x_{k,p}^t$. This constraint acts like a trust region. Figure 1 shows the steps of this construction.

In this work, we solve a relaxation of this constrained model as explained in (Öztoprak & Birbil, 2018, Section 2.2). The minimum value of the relaxed model is attained at the point $x_{k,p} + s_{k,p}$ with

$$s_{k,p} = c_{g,p}(\delta) g_{k,p} + c_{y,p}(\delta) y_{k,p} + c_{s,p}(\delta) s_{k,p}^t, \tag{4}$$

where $y_{k,p} := g_{k,p}^t - g_{k,p}$. Here, the coefficients are given as

$$c_{g,p}(\delta) = -\frac{\|s_{k,p}^t\|^2}{2\delta}, \quad c_{y,p}(\delta) = -\frac{\|s_{k,p}^t\|^2}{2\delta\theta}[-(y_{k,p}^\top s_{k,p}^t + 2\delta)(s_{k,p}^t)^\top g_{k,p} + \|s_{k,p}^t\|^2 y_{k,p}^\top g_{k,p}],$$

$$c_{s,p}(\delta) = -\frac{\|s_{k,p}^t\|^2}{2\delta\theta}[-(y_{k,p}^\top s_{k,p}^t + 2\delta)y_{k,p}^\top g_{k,p} + \|y_{k,p}\|^2 (s_{k,p}^t)^\top g_{k,p}],$$

with

$$\theta = \left(y_{k,p}^\top s_{k,p}^t + 2\delta\right)^2 - \|s_{k,p}^t\|^2 \|y_{k,p}\|^2 \quad \text{and} \quad \delta = \frac{1}{2}\left(\|s_{k,p}^t\|\left(\|y_{k,p}\| + \frac{1}{\eta}\|g_{k,p}\|\right) - y_{k,p}^\top s_{k,p}^t\right), \tag{5}$$

where $0 < \eta < 1$ is a constant which controls the size of $s_{k,p}$ by imposing the condition $\|s_{k,p}\| \leq \eta\|s_{k,p}^t\|$. Then, the adaptive model step becomes $s_k = (s_{k,p_1}, \ldots, s_{k,p_n})$. We note that our construction in terms of different parameter groups lends itself to constructing a different model for each parameter subspace.

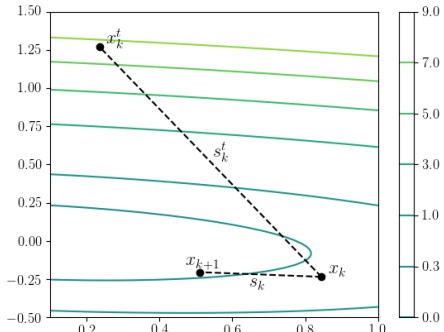

Figure 1: An iteration of SMB on a simple quadratic function. For simplicity we assume that there is only one parameter group, and hence, we drop the subscript $p$. The algorithm first computes the trial point $x_k^t$ by taking the stochastic gradient step $s_k^t$. If this point is not acceptable, then it builds a model using the information at $x_k$ and $x_k^t$, and computes the next iterate $x_{k+1} = x_k + s_k$. Note that $s_k$ not only have a smaller length compared to the trial step $s_k^t$, but it also lies along a direction decreasing the function value.

We summarize the steps of SMB in Algorithm 1. Line 5 shows the trial point, which is obtained with the standard stochastic gradient step. If this step satisfies the stochastic Armijo condition, then we proceed with the next iteration (line 8). Otherwise, we continue with bulding the models for each parameter group (lines 10- 12), and move to the next iteration with the model building step in line 13.

---

**Algorithm 1:** SMB: Stochastic Model Building

1   **Input:** $x_1 \in \mathbb{R}^n$, stepsizes $\{\alpha_k\}_{k=1}^T$, mini-batch sizes $\{m_k\}_{k=1}^T$, $c > 0$, and $\alpha_{max}$ satisfying (8)
2   **for** $k = 1, \ldots, T$ **do**
3     $f_k = f(x_k, \xi_k)$, $g_k = \frac{1}{m_k} \sum_{i=1}^{m_k} g(x_k, \xi_{k,i})$;
4     $s_k^t = -\alpha_k g_k$;
5     $x_k^t = x_k + s_k^t$;
6     $f_k^t = f(x_k^t, \xi_k)$, $g_k^t = \frac{1}{m_k} \sum_{i=1}^{m_k} g(x_k^t, \xi_{k,i})$;
7     **if** $f_k^t \leq f_k - c\,\alpha_k \|g_k\|^2$ **then**
8       $x_{k+1} = x_k^t$ ;
9     **else**
10       **for** $p = 1, \ldots, n$ **do**
11         $y_{k,p} = g_{k,p}^t - g_{k,p}$;
12         $s_{k,p} = c_{g,p}(\delta)g_{k,p} + c_{y,p}(\delta)y_{k,p} + c_{s,p}(\delta)s_{k,p}^t$;
13       $x_{k+1} = x_k + s_k$ with $s_k = (s_{k,p_1}, \ldots, s_{k,p_n})$;

---

## 3   CONVERGENCE ANALYSIS

The steps of SMB can be considered as a special quasi-Newton update:

$$x_{k+1} = x_k - \alpha_k H_k g_k, \qquad (6)$$

where $H_k$ is a symmetric positive definite matrix as an approximation to the inverse Hessian matrix. In Appendix A.1, we explain this connection and give an explicit formula for the matrix $H_k$. We also prove that there exists $\underline{\kappa}, \overline{\kappa} > 0$ such that for all $k$ the matrix $H_k$ satisfies

$$\underline{\kappa}I \preceq H_k \preceq \overline{\kappa}I, \qquad (7)$$

where for two matrices $A$ and $B$, $A \preceq B$ means $B - A$ is positive semidefinite. It is important to note that $H_k$ is built with the information collected around $x_k$, particularly, $g_k$. Therefore, unlike stochastic quasi-Newton methods, $H_k$ is correlated with $g_k$, and hence, $\mathbb{E}[H_k g_k]$ is very difficult to analyze. Unfortunately, this difficulty prevents us from using the general framework given by Wang et al. (2017).

To overcome this difficulty and carry on with the convergence analysis, we modify Algorithm 1 such that $H_k$ is calculated with a new independent mini batch, and therefore, it is independent of $g_k$. By doing so, we still build a model using the information around $x_k$. Assuming that $g_k$ is an unbiased estimator of $\nabla f$, we conclude that $\mathbb{E}[H_k g_k] = H_k \nabla f$. In the rest of this section, we provide a convergence analysis for this modified algorithm which we will call as SMBi (i for independent batch). The steps of SMBi are given in Algorithm 2. As Step 11 shows, we obtain the model building step with a new random batch.

---

**Algorithm 2:** SMBi: $H_k$ with an independent batch

---

1 **Input:** $x_1 \in \mathbb{R}^n$, stepsizes $\{\alpha_k\}_{k=1}^T$, mini-batch sizes $\{m_k\}_{k=1}^T$, $c > 0$, and $\alpha_{max}$ satisfying (8)

2 **for** $k = 1, \ldots, T$ **do**

3 $\quad$ $f_k = f(x_k, \xi_k)$, $g_k = \frac{1}{m_k} \sum_{i=1}^{m_k} g(x_k, \xi_{k,i})$;

4 $\quad$ $s_k^t = -\alpha_k g_k$;

5 $\quad$ $x_k^t = x_k + s_k^t$;

6 $\quad$ $f_k^t = f(x_k^t, \xi_k)$, $g_k^t = \frac{1}{m_k} \sum_{i=1}^{m_k} g(x_k^t, \xi_{k,i})$;

7 $\quad$ **if** $f_k^t \le f_k - c \, \alpha_k \|g_k\|^2$ **then**

8 $\quad\quad$ $x_{k+1} = x_k^t$ ;

9 $\quad$ **else**

10 $\quad\quad$ **for** $p = 1, \ldots, n$ **do**

11 $\quad\quad\quad$ Choose a new independent random batch $\xi_k'$;

12 $\quad\quad\quad$ $g_k' = \frac{1}{m_k} \sum_{i=1}^{m_k} g(x_k, \xi_{k,i}')$;

13 $\quad\quad\quad$ $(s_k^t)' = -\alpha_k g_k'$, $(x_k^t)' = x_k + (s_k^t)'$;

14 $\quad\quad\quad$ $(g_k^t)' = \frac{1}{m_k} \sum_{i=1}^{m_k} g((x_k^t)', \xi_{k,i}')$, $y_{k,p}' = (g_{k,p}^t)' - g_{k,p}'$;

15 $\quad\quad\quad$ $s_{k,p} = -\alpha_k H_{k,p}' g_k$, where $H_{k,p}'$ is calculated using $g_k'$ and $y_k'$ as defined in Appendix A.1;

16 $\quad\quad$ $x_{k+1} = x_k + s_k$ with $s_k = (s_{k,1}, \ldots, s_{k,n})$;

---

**Assumptions:** Before providing the analysis, let us assume that $f : \mathbb{R}^n \to \mathbb{R}$ is continuously differentiable, lower bounded by $f^{low}$, and there exists $L > 0$ such that for any $x, y \in \mathbb{R}^n$, $\|\nabla f(x) - \nabla f(y)\| \le L \|x - y\|$. We also assume that $\xi_k$, $k \ge 1$, are independent samples and for any iteration $k$, $\xi_k$ is independent of $\{x_j\}_{j=1}^k$, $\mathbb{E}_{\xi_k}[g(x_k, \xi_k)] = \nabla f(x_k)$ and $\mathbb{E}_{\xi_k}[\|g(x_k, \xi_k) - \nabla f(x_k)\|^2] \le \sigma^2$, for some $\sigma > 0$.

In order to be in line with practical implementations and with our experiments, we first provide an analysis covering the constant stepsize case for (possibly) non-convex objective functions.

Below, we denote by $\xi_{[T]} = (\xi_1, \ldots, \xi_T)$ the random samplings in the first $T$ iterations. Let $\alpha_{max}$ be the maximum stepsize that is allowed in the implementation of SMBi with

$$\alpha_{max} \ge \frac{-1 + \sqrt{1 + 16\eta^2}}{4L\eta}. \tag{8}$$

This hyper-parameter of maximum stepsize is needed in the theoretical results. The same parameter can also be used to apply automatic stepsize adjustment (see our numerical experiments with stepsize auto-scheduling in Section 4.2). Observe that since $\eta^{-1} > 1$, assuming $L \ge 1$ implies that it suffices to choose $\alpha_{max} \ge 1$ to satisfy (8). The proof of the following convergence result is given in Appendix A.1.

**Theorem 1.** *Suppose that our assumptions above hold and $\{x_k\}$ is generated by SMBi as given in Algorithm 2. Suppose also that $\{\alpha_k\}$ in Algorithm 2 satisfies that $0 < \alpha_k < 2/(L\eta^{-1} + 2L^2\alpha_{max}) \le$*

$\alpha_{max}$ *for all $k$. For given $T$, $R$ be a random variable with the probability mass function*

$$\mathbb{P}_R(k) := \mathbb{P}\{R = k\} = \frac{\alpha_k/(\eta^{-1} + 2L\alpha_{max}) - \alpha_k^2 L/2}{\sum_{k=1}^T (\alpha_k/(\eta^{-1} + 2L\alpha_{max}) - \alpha_k^2 L/2)},$$

*for $k = 1, \ldots, T$. Then, we have*

$$\mathbb{E}[\|\nabla f(x_R)\|^2] \leq \frac{D_f + (\sigma^2 L/2) \sum_{k=1}^T (\alpha_k^2/m_k)}{\sum_{k=1}^T (\alpha_k/(\eta^{-1} + 2L\alpha_{max}) - \alpha_k^2 L/2)},$$

*where $D_f := f(x_1) - f^{low}$ and the expectation is taken with respect to $R$ and $\xi_{[T]}$. Moreover, if we choose $\alpha_k = 1/(L\eta^{-1} + 2L^2\alpha_{max})$ and $m_k = m$ for all $k = 1, \ldots, T$, then this reduces to*

$$\mathbb{E}[\|\nabla f(x_R)\|^2] \leq \frac{2L(\eta^{-1} + 2L\alpha_{max})^2 D_f}{T} + \frac{\sigma^2}{m}.$$

Using this theorem, it is possible to deduce that stochastic first-order oracle complexity of SMB with random output and constant stepsize is $\mathcal{O}(\epsilon^{-2})$ (Wang et al., 2017, Corollary 2.12).

In (Wang et al., 2017, Theorem 2.5), it is shown that under our assumptions above and the extra assumption of $0 < \alpha_k \leq \frac{1}{L(\eta^{-1} + 2L\alpha_{max})} \leq \alpha_{max}$, if the point sequence $\{x_k\}$ is generated by SMBi method (when $H_k$ is calculated by an independent batch in each step) with batch size $m_k = m$ for all $k$, then there exists a positive constant $M_f$ such that $\mathbb{E}[f(x_k)] \leq M_f$. Using this observation, the proof of Theorem 1, and Theorem 2.8 in (Wang et al., 2017), we can also give the following complexity result when the stepsize sequence is diminishing for non-convex objective functions.

**Theorem 2.** *Let the batch size be $m$ and assume that $\alpha_k = \frac{1}{L(\eta^{-1} + 2L\alpha_{max})} k^{-\phi}$ with $\phi \in (0.5, 1)$ for all $k$. Then $\{x_k\}$ generated by SMBi satisfies that*

$$\frac{1}{T} \sum_{k=1}^T \mathbb{E}[\|\nabla f(x_k)\|^2] \leq 2L(\eta^{-1} + 2L\alpha_{max})(M_f - f^{low})T^{\phi-1} + \frac{\sigma^2}{(1-\phi)m}(T^{-\phi} - T^{-1})$$

*for some $M_f > 0$, where $T$ denotes the iteration number. Moreover, for a given $\epsilon \in (0, 1)$, to guarantee that $\frac{1}{T} \sum_{k=1}^T \mathbb{E}[\|\nabla f(x_k)\|^2] < \epsilon$, the number of iterations $T$ needed is at most $O\left(\epsilon^{-\frac{1}{1-\phi}}\right)$.*

We are now ready to assess the performance of SMB and SMBi with some numerical experiments.

## 4 NUMERICAL EXPERIMENTS

In this section, we compare SMB and SMBi against SGD, Adam Kingma & Ba (2015), and SLS (SGD+Armijo) Vaswani et al. (2019). We have chosen SLS since it is a recent method that uses stochastic line search with backtracking. We have conducted experiments on multi-class classification problems using neural network models[1]. The codes to conduct our experiments are given in the supplementary file.

### 4.1 CONSTANT STEPSIZE

We start our experiments with constant stepsizes for all methods. We should point out that SLS method adjusts the stepsize after each backtracking process and also uses a stepsize reset algorithm between epochs. We refer to this routine as stepsize auto-scheduling. Therefore, we find it unfair to compare SLS with other methods with constant stepsize. Please, see Section 4.2 for a discussion about stepsize auto-scheduling using SMB.

**MNIST dataset.** On the MNIST dataset, we have used the one hidden-layer multi-layer perceptron (MLP) of width 1,000. We compare all methods after cross-validating their best performances from the set of learning rates, $\{0.001, 0.01, 0.1, 0.25, 0.5, 0.75, 1.0\}$. For SMB and SLS, we have

---

[1]The implementations of the models are taken from https://github.com/IssamLaradji/sls

used the default hyper-parameter value $c = 0.1$ of SLS that appears in the Armijo condition (also recommended by the authors of SLS)

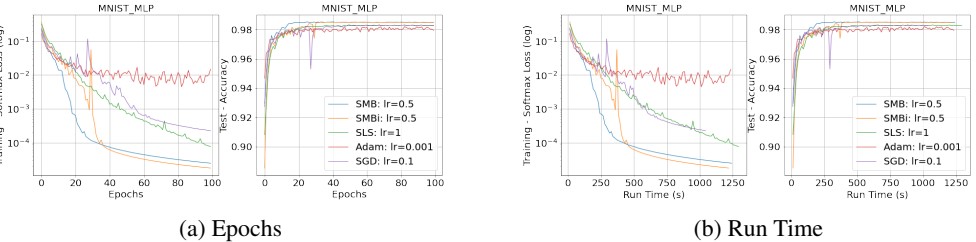

(a) Epochs  (b) Run Time

Figure 2: Classification on MNIST with an MLP model.

In Figure 2, we see the best performances of all five methods on the MNIST dataset with respect to epochs and run time. The reported experiments consist of five independent runs where results are averaged. Even though SMB and SMBi may calculate an extra function value (forward pass) and a gradient (backward pass), we see in this problem that SMB and SMBi achieve the best performance with respect to the run time as well as the number of epochs. More importantly, the generalization performances of SMB and SMBi are also better than the remaining three methods.

It should be pointed out that, in practice, choosing a new independent batch means the SMBi method can construct a model step in two iteration using two batches. This way the computation cost for each iteration is reduced but the model steps can only be taken in half of the iterations in the epoch. As seen in Figure 2, this does not seem to effect the performance significantly.

**CIFAR10 and CIFAR100 datasets.**   For the CIFAR10 and CIFAR100 datasets, we have used the standard image-classification architectures ResNet-34 (He et al., 2016) and DenseNet-121 (Huang et al., 2017) . Due to the high computational costs of these architectures, we report the results of a single run of each method. For, Adam we have used the default learning rate 0.001, and for SGD, we have set the tuned learning rate to 0.1 as reported in Vaswani et al. (2019). For SMB and SLS, we have again used the default learning rate of 1.0 and Armijo constant $c = 0.1$ of SLS.

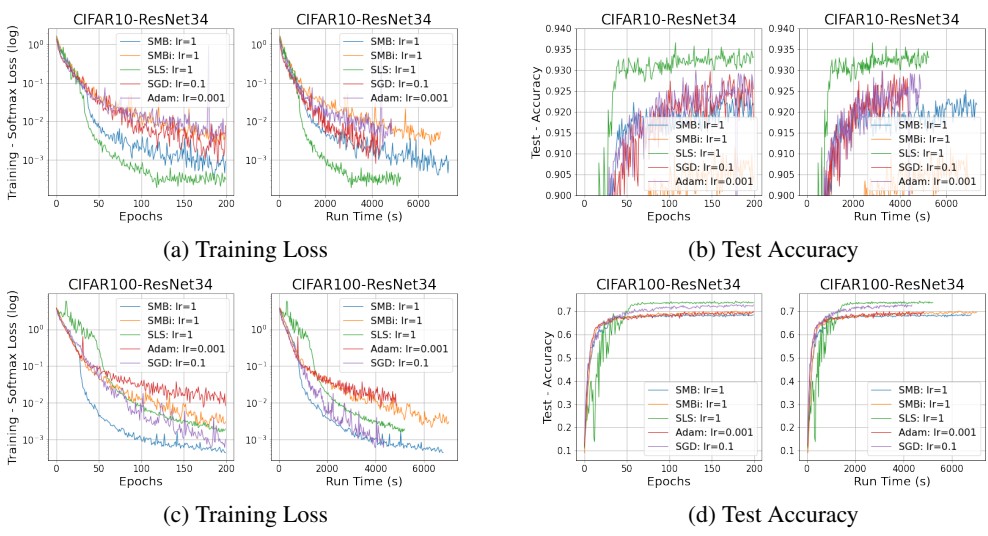

(a) Training Loss  (b) Test Accuracy

(c) Training Loss  (d) Test Accuracy

Figure 3: Classification on CIFAR10 (a, b) and CIFAR100 (c, d) with ResNet-34 model.

In Figure 3, we see that on CIFAR10-Resnet34 and CIFAR100-Resnet34, SMB performs better than Adam and SGD algorithms. However, its performance is only comparable to SLS. Even though SMB reaches a lower loss function value in CIFAR100-Resnet34, this advantage does not show in test

accuracy. As mentioned in the beginning of this section, SLS method adjusts the stepsize after each backtracking process and, in order to prevent diminishing stepsizes, it uses a stepsize reset algorithm between epochs. SMB does not benefit from this kind of stepsize auto-scheduling. We will define an auto-scheduling for SMB stepsizes in Section 4.3 so that we obtain a fairer comparison between SMB and SLS.

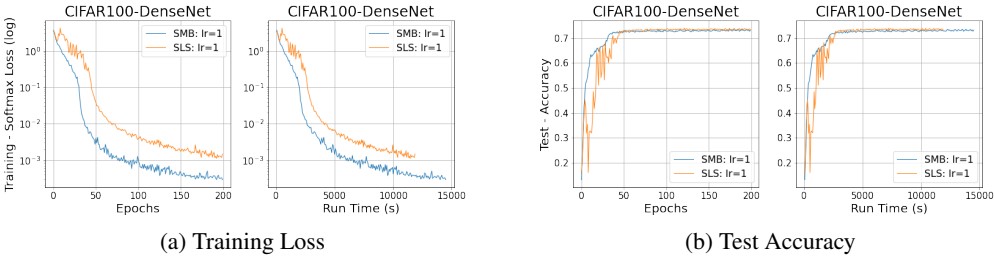

(a) Training Loss                              (b) Test Accuracy

Figure 4: Classification on CIFAR100 with DenseNet-121 model.

In Figure 4, we see a comparison of performances of SMB and SLS on CIFAR100-DenseNet121. SMB with a constant stepsize outperforms SLS on train loss and reaches to high test accuracy before SLS. Vaswani et al. (2019) show that SLS with these settings outperforms Adam and SGD on this problem both in terms of traning loss and test accuracy.

## 4.2 STEPSIZE AUTO-SCHEDULING

As expected SMB can take many model steps, when learning rate is too large. Then, extra mini-batch function and gradient evaluations can slow down the algorithm (*c.f.*, Figure 3). We believe that the number of model steps taken in an epoch (when the Armijo condition is not satisfied) can be a good measure to adjust the learning rate in the next epoch. This can lead to an automatic learning rate scheduling algorithm. We did preliminary experiments with a simple stepsize auto-scheduling routine, The results are given in Figure 5. At the end of each epoch, we multiply the stepsize by 0.9 when the model steps taken in an epoch is more than 5% of the total steps taken. Otherwise, we divide the stepsize by 0.9, unless the division ends up with a stepsize greater than the maximum stepsize allowed, $\alpha_{max}$. The value 0.9 is the backtracking ratio of SLS and we consider 5% as a hyper-parameter. Figure 5 shows, on the training loss, that both SMB and SMBi perform better than the other methods. For the test accuracy, SMB performs better than all other methods, and SMBi performs comparable to SLS.

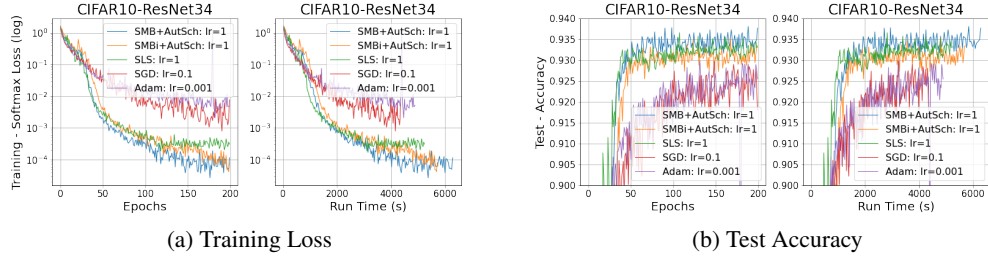

(a) Training Loss                              (b) Test Accuracy

Figure 5: Performances of SMB and SMBi with auto-scheduled stepsizes on CIFAR10.

## 4.3 ROBUSTNESS WITH RESPECT TO STEPSIZE

Our last set of experiments are devoted to demonstrating the robustness of SMB. The preliminary results in Figure 6 show that SMB is more robust to the choice of the learning rate than Adam and SGD, especially in deep neural networks. This aspect of SMB needs more attention theoretically and experimentally.

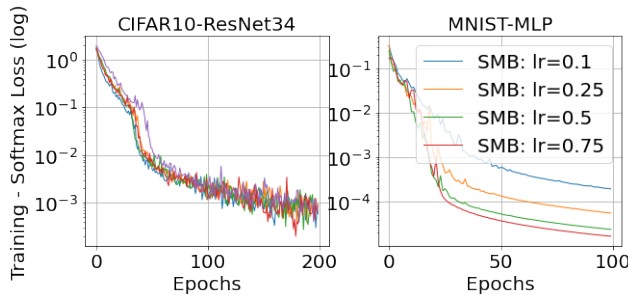

Figure 6: Robustness of SMB under different choices of the learning rate.

## 5 CONCLUSION

SMB is a fast alternative to stochastic gradient method. The algorithm provides a model building approach that replaces the one-step backtracking in stochastic line search methods. We have analyzed the convergence properties of a modification of SMB by rewriting its model building step as a quasi-Newton update and constructing the scaling matrix with a new independent batch. Our numerical results have shown that SMB converges fast and its performance is much more insensitive to the selected stepsize than Adam and SGD algorithms. In its current state, SMB lacks any internal learning rate adjusting mechanism that could reset the learning rate depending on the progression of the iterations. As shown in Section 4.3, SMB can greatly benefit from a stepsize auto-scheduling routine. This is a future work that we will consider.

**Limitations.** Our convergence rate analysis is given for the alternative algorithm SMBi which can perform well agains other methods but consistently underperforms the original SMB method. This begs for a convergence analysis for the SMB method.

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
