# OpenReview forum: "Bolstering Stochastic Gradient Descent with Model Building"
_ICLR.cc/2022/Conference — ICLR 2022 Submitted_

### Official Review · Reviewer_4z74 · 2021-10-28

**Correctness:** 3
**Technical Novelty And Significance:** 2
**Empirical Novelty And Significance:** 2
**Recommendation:** 5
**Confidence:** 4

**Main Review:**

**Strengths**
The proposed method SMB/SMBi has several advantages. First of all, unlike general preconditioned stochastic gradient methods, the preconditioner here is a blockwise diagonal matrix where each block corresponds to a parameter group in the deep neural network. In addition, because the preconditioned update is only expressed so to facilitate theoretical analysis, the actual implementation itself does not involve computing or storing matrices at all --- only vector inner products are involved in each parameter group update. This is particularly suitable for implementation in popular deep learning libraries like PyTorch where parameters are stored in groups.

SMB also does not use backtracking as is typical in line search with the Armijo condition. When the condition is failed for the stochastic gradient direction, SMB simply computes a new direction based on a stochastic "quasi-Newton" update, and take a step along this direction instead. This lets SMB avoid potentially doing many additional forward passes which can be expensive.

Finally, the empirical results show that by using additional simple heuristics to adjust the step size based on how often the Armijo condition has failed in the previous epoch, SMB can achieve state-of-the-art performance on CIFAR10 with a ResNet34 architecture, both in terms of the training loss and test accuracy.


**Weaknesses**
The main concerns I have with this paper are:
1. Although convergence results are given in the non-convex setting, the analysis is rather trivial. The crux of the proofs for the theorems is to first re-write the SMB/SMBi update as a preconditioned stochastic gradient step $x_{k+1} = x_k - \alpha_k H_k g_k$, and this part comes directly from the work of Oztoprak and Birbil (2018) except here the deterministic gradients are replaced with their stochastic counterparts. Then the proof boils down to bounding the extreme eigenvalues of each of the diagonal blocks in $H_k$. This part of the analysis is also nearly identical to what's in Oztoprak and Birbil (2018). Finally, with the extreme eigenvalues at hand, the non-convex results given by Wang et al. (2017) can be directly invoked to obtain Theorem 1 and 2 in this paper. The theoretical analysis therefore is simple as previous works have done all the heavy-lifting,  but at the same time I consider the contribution to be neither novel nor significant.
2. Because the algorithm is directly adapted from that of Oztoprak and Birbil (2018), the preconditioner $H_k$ would be correlated with the stochastic gradient. This would not fit into the analysis of Wang et al. (2017) as their theory requires $H_k$ to be independent of $\nabla f(x_k,\xi_k)$ conditioned on the past. The algorithm in Wang et al. (2017) constructs $H_k$  based on a truncated history of the past stochastic gradients and therefore is independent of the randomness in iteration $k$. To circumvent this issue, the authors here suggest to analyze a different version of SMB, which uses an independent minibatch to compute $H_k$, and the variant is named SMBi. Although this lets the analysis go through, it inevitably introduces a *gap between theory and practice*, since the experiments demonstrate that SMB almost always perform better than SMBi. This observation makes sense as although we are using stochastic quantities to construct the model, the model can be seen as a locally approximation to the minibatch objective, while using independent batches to compute the model does not seem to make a lot of sense.
3. Figure 1: Although it is true that in this particular case (seed, rather), "$s_k$  has a smaller length compared to the trial step $s_k^t$, the step we would have taken with $s_k^t$ is usually scaled with a step size instead of the unit step size taken by $s_k$ in SMB. More importantly, the direction obtained in $s_k^t$, the stochastic gradient step depends on which feature vector $\xi_k$ is sampled, and so it shouldn't always be the case that $s_k^t$ "lies along a direction decreasing the function value" while $s_k$ does not, unless this can be proven otherwise. For these reasons I find this illustration slightly misleading.
4. Robustness to step size: assuming the experiments in Figure 6 are using constant step sizes, it is unclear to me what the point of robustness to step size means given that SMB/SMBi only works well in general when step size auto-scheduling is required. If the authors indeed recommend the step size heuristic in Section 4 to be adopted in practice, then I see little value in this robustness experiment, but I could be interpreting the experiment incorrectly. Instead, I would recommend a robustness experiment on the other hyperparameters such as the percentage threshold mentioned in Section 4 and the step size multiplication factor. The last issue I have with Figure 6 is it seems like for MNIST-MLP, different step size can lead to an order of magnitude difference in the final training loss. Does that imply SMB isn't very robust to the step size on this dataset/model combination?
5. It would be great to see a replication of the auto-scheduling experiment on CIFAR100-ResNet34 as well, especially to see whether with the auto-scheduling, the "converge fast but generalize worse" observation in Figure 3 (c,d) can be mitigated.

**Minor concerns / typos**
- In the objective (Equations (1) and (2)), the dimensionality of the problem is $n$; however, $n$ is later also used to denote the number of parameter groups at the top of page 3. The latter implies the overall dimensionality of the problem should be $\sum_{j=1}^n p_j$ where $p_j$ is the number of parameters in group $j$.
- It's unclear to me why $f_k=f(x_k,\xi_k)$ is not expressed as the sum of individual function values of a mini-batch, as you do for the stochastic gradients $g_k$. I suppose it should be and this is just to avoid cluttering, or are the $f_k$'s computed in some other way?
- Top of page 5: when concluding that $\mathbb{E}[H_kg_k] = H_k\nabla f$, one should be more explicit that the expectation is taken over $\xi_k$, the first minibatch sampled at that iteration, excluding the randomness in $H_k$, since $H_k$ is also a random quantity it's just that its randomness comes from a separate, independent minibatch.
- Top of page 6: "For given $T$, $R$ be a random variable" -> "For *a* given $T$, *let* $R$ be a random variable"
- Proof of Theorem 1 (page 11, section A.1):
	- First line: "can be expressed a special" -> "can be expressed *as* a special"
	- Please use a different notation for $\sigma=(y_k^\top s_k^t + 2\delta)^2 - \|s_k^t\|^2 \|y_k\|^2$ as $\sigma$ is already used in Assumption $\mathbb{E}_{\xi_k}[\|g(x_k,\xi_k)-\nabla f(x_k)\|^2] \leq \sigma^2$.
- Proof of Theorem 1 (page 12, section A.1):
	- Second equality in upper bounding $\lambda_{\max}$: the last $+y_{k,p}^\top g_{k,p}$ should be a minus instead. (typo)
- It would be helpful to further explain the motivation behind the *stochastic model building* part of the algorithm, rather than just citing away Oztoprak and Birbil (2018). In particular, a justification of why their globalization strategy should also work well in the stochastic setting would significantly strengthen the paper.

---
**References mentioned**:
Oztoprak and Birbil (2018): An alternative globalization strategy for unconstrained optimization
Wang et al. (2017): Stochastic Quasi-Newton Methods for Nonconvex Stochastic Optimization
Nocedal and Wright (2006): Numerical Optimization
Vaswani et al. (2019) : Painless Stochastic Gradient: Interpolation, Line-Search, and Convergence Rates


**Summary Of The Paper:**

The authors proposed a method called stochastic model building (SMB) that uses a combination of existing techniques to get faster convergence in stochastic non-convex optimization. In particular, they use a stochastic adaptation of the model-building globalization strategy from Oztoprak and Birbil (2018), in which the deterministic Armijo condition check is also computed using stochastic gradients. By re-writing their update as a preconditioned SGD step, they are able to bound the spectrum of the preconditioner. This allows them to obtain convergence results for smooth and non-convex objectives by directly invoking the analysis of Wang et al. (2017). Experiments demonstrate that with additional heuristics, their proposed method can outperform state-of-the-art optimizers on common deep learning benchmarks.

**Summary Of The Review:**

Although the paper proposes an interesting, easy-to-implement algorithm for non-convex stochastic optimization, I am hesitant to recommend for acceptance at this point for its lack of significantly novel contribution and some confusing aspects of the experiments. I am willing to increase my score if my concerns are well addressed.

---

> ### Author Response · Authors · 2021-11-18
> **Reply**
>
> We thank the reviewer for the valuable feedback and suggestions. We would like to clarify the following points:
>
> The trial step is given as $s_k^t = \alpha_k g_k$ where $\alpha_k$ is the step size and $g_k$ is the stochastic gradient. Therefore, the model step $s_k$ involves the step size $\alpha_k$ implicitly and it is always true that $||s_k|| < ||s_k^t|| = |\alpha_k| ||g_k||$.
>
> We agree with the reviewer that neither $s_k^t$ nor $s_k$ has to lie along a direction decreasing the function value. The step $s_k$ provides a decrease only for the model function and a decrease in expectation for the objective function. We will clarify this in the explanation of Figure 1.
>
> The robustness experiments were given for the fixed step size cases where no auto-scheduling was used. We will provide robustness experiments on the other hyper-parameters in the next version of the paper as the reviewer suggests. It is true that SMB is less robust to the step size in the MNIST-MLP problem than it is in the other problems but it converges to a good solution on a much bigger interval of step sizes than Adam and SGD as our experiments show. We can add these experiments to the supplementary material where they may fit in the best.

---

### Official Review · Reviewer_8Xqy · 2021-11-02

**Correctness:** 4
**Technical Novelty And Significance:** 2
**Empirical Novelty And Significance:** Not applicable
**Recommendation:** 5
**Confidence:** 4

**Main Review:**

1- The paper is an incremental work and is mainly a stochastic variant of Öztoprak et al 2018. This paper doesn't offer anything technical since all the theoretical results are just slight modifications of already existing results. On the other hand, the empirical results also don't show a big advancement.

2-  Generally tuning both direction and step size is not a new approach. For example, the diagonal variant of AdaGrad has a similar approach. Also in the proposed algorithm in each step, it can require a matrix-vector multiplication and it cannot be relaxed. Parameter partitioning is a way to make the matrix block-diagonal and to reduce the matrix-vector multiplication. However, in the paper, it is not clarified why partitioning is useful.

3- The paper claims that unlike line search there is no backtracking step. However, to set \eta their proposed algorithm needs backtracking.

4- It hasn’t been mentioned in the paper how many times each experiment ran? It would be more reliable if each experiment runs several times and the average of them is plotted.


**Summary Of The Paper:**


This paper proposes an alternative to stochastic line search which is based on forwarding step model building which corrects the direction of move and its magnitude at the same time. In its proposed algorithm it first checks if the given step size satisfies the stochastic line search. If yes then just use the step size and do the SGD update. Otherwise, it builds linear models around two points and combines these two models and minimizes this new model and this becomes the new iterate value.



**Summary Of The Review:**

The paper is an incremental work and is mainly a stochastic variant of Öztoprak et al 2018. This paper doesn't offer anything technical since all the theoretical results are just slight modifications of already existing results. On the other hand, the empirical results also don't show a big advancement.

---

> ### Author Response · Authors · 2021-11-17
> **Reply**
>
> We thank the reviewer for the valuable feedback and suggestions. We would like to clarify the following points:
>
> The description of the SMB steps as a matrix multiplication is only to utilize the theoretical framework of quasi-Newton methods. The steps are actually computed only with inner products and no matrix multiplication is involved.
>
> In all of the experiments $\eta$ is fixed and set to 0.99. The algorithm seems to be robust to the choice of $\eta$. We use no backtracking  in the SMB algorithm, just calculate the model minimum once when the Armijo condition is not satisfied.
>
> All the experiments, except in the MNIST-MLP problem, ran only once. We will repeat the experiments several times in the next version of the paper.

---

### Official Review · Reviewer_hEnG · 2021-11-02

**Correctness:** 3
**Technical Novelty And Significance:** 2
**Empirical Novelty And Significance:** 1
**Recommendation:** 3
**Confidence:** 3

**Main Review:**

Strengths
There is some novelty in the proposed algorithm, and the paper has made some efforts to extend the model building approach to the stochastic setting.

Weakness

1. From theoretical side, by enforcing a relatively strong independence batch assumption, the convergence analysis is not a surprise given the reference Wang 2017. However, the experiments shows that SMB  consistently outperforms SMBi, it seems to me that dependence is the key for getting better performance.  Hence, it is unsatisfactory if the convergence property of SMB is not demystified and completely ignored.

2. Since the paper has targeted its proposed algorithm a special case of quasi-newton method (e.g. (6)), it would be unsatisfactory if comparison with SQN is not made.

3. The experiment conclusion is based on a single run, it is not convicing to me that the reported advantage of SMB is as reliable as described. I would recommend spending more time effort in the experiments before submission.

4 Figure 3 shows that SLS outperforms SGD and ADAM while SGD and ADAM outperform SMB and SMBi. The last figure shows that SMB with Auto-scheduled stepsize performs slightly better than SLS. What is the most important factor in getting good performance? Would the benefits mainly come from the Auto-scheduled stepsize? Can we equip SGD or ADAM with similar scheduled stepsize? This appears to be very promising.


Some minor issues.
1. \mathbb{E}[F(x; \xi )] is deﬁned at the beginning but never used.

2. It is confusing that $n$ is used to deﬁne both the number of parameter blocks and the dimension of X.

3. What is  $f_{k,p}$ defined in page 3?



**Summary Of The Paper:**

The paper presents a stochastic version of the model building algorithm (SMB)  by taking the subsequent iterate to be the minimizer of a quadratic model build from two iterates (the current and the one based on conventional SGD). Empirical results suggest that the proposed SMB has some great generalization performance.

**Summary Of The Review:**

While this paper makes some interesting contribution of a new stochastic algorithm for deep learning, it appears to be far from a complete work. I think the paper needs to be substantially improved in both theory and empirical study.

---

> ### Author Response · Authors · 2021-11-17
> **Reply**
>
> We thank the reviewer for the valuable feedback and suggestions. We would like to mention that in our experiments SGD did not benefit from the same auto-schedule mechanism as SMB did. We agree that the good results of auto-scheduled SMB needs to be studied in more detail.

---

### Official Review · Reviewer_xM1f · 2021-11-02

**Correctness:** 3
**Technical Novelty And Significance:** 2
**Empirical Novelty And Significance:** 2
**Recommendation:** 5
**Confidence:** 5

**Main Review:**

1- The majority of the manuscript is well-written and easy to understand. However, some parts require further explanation and clarification that the reviewer explains in the following comments.

2- The literature part needs to be strengthened.

3- The point that is mentioned in Page 2 "It also incorporates the most recent curvature information from the current point. This is in contrast with the stochastic quasi-Newton methods which use the information from the previous steps" is $\textbf{not}$ novel! The studies [1] and [2] (to name a few) introduced this concept earlier.

4- Does the inequality (3) guarantee a decrease in the main objective function or just the stochastic objective function? If just the stochastic objective function, how can we trust the step length?

5- In Page 3 (line 3 in the second paragraph) the authors mentioned group $p$ which is not introduced before. The text needs to be more clear and readable.

6- One of the main goals of this paper is related to reducing the computational costs of tuning. However, in the proposed algorithms 1 and 2, there are many hyper-parameters that still requires tuning. The authors need to exactly mention how they claim the idea of reducing tuning attempts.

7- The theoretical results are quite basic and more stronger results are expected. Theorem 1 does not guarantee any convergence, it just shows that the norm of gradient (in expectation) is bounded above by some terms that are NOT diminishing as the algorithm progresses. If $\alpha_k = \mathcal{O}(L^{-2})$ (which is super tiny), the proposed method converges to the neighborhood of stationary point. Even with diminishing step sizes, the proved convergence result does not guarantee any convergence to a stationary point (just the neighborhood around a stationary point). All in all, the theoretical results definitely require modification.

8- Do the authors consider the extra cost per each iteration in the plots with $x$-axis as $\textbf{Epochs}$? Also, instead of averaging over 5 runs, it is better to show the error band. The mini-batch size is not reported for the experiments, it is very important hyper-parameter as well.

9- That is good that the authors provided some results regarding the robustness of their method.

10- In the abstract, it is mentioned that $\textbf{This novel
diagonalization approach makes the selected step lengths adaptive.}$ The proposed method is not adaptive in terms of step size! In both algorithms 1 and 2, there is not any adaptive rule for the step size.


[1] Gao, W., & Goldfarb, D. (2018). Block BFGS methods. SIAM Journal on Optimization, 28(2), 1205-1231.

[2] Berahas, A. S., Jahani, M., Richtárik, P., & Takáč, M. (2020). Quasi-newton methods for deep learning: Forget the past, just sample.

**Summary Of The Paper:**

This paper presents a model building approach that replaces the one-step backtracking in stochastic line search methods. The authors provided some convergence guarantees for their proposed method and evaluated their approach on some image classification tasks.

**Summary Of The Review:**

The theoretical results needs to be more strengthened. The contribution is ok but not good enough.

---

> ### Author Response · Authors · 2021-11-17
> **Reply**
>
> We thank the reviewer for the valuable feedback and suggestions. We would like to clarify the following points:
>
> The decrease guaranteed in equation (3) is only for the stochastic objective function.  There are no guarantees for the step length to work on any individual step but only in expectation.
>
> In majority of the experiments, we have used the hyper-parameters with their default values (except in the MNIST-MLP in which we have used the learning rate of 0.5 as opposed to 1). We have observed that SMB is not only robust with respect to learning rate but also to the other hyper-parameters. As another reviewer suggested, we plan to add more experiments on robustness with respect to other hyper-parameters in the next version of the paper.
>
> In order to consider the extra cost per each iteration, we provided run-time plots as well as plots with respect to epochs.

---

### Decision · Program_Chairs · 2022-01-20

**Decision:**

Reject

**Comment:**

There was a consensus among the reviewers to reject the paper. While they noted that the paper proposed a new interesting stochastic algorithm for deep learning, they think the paper needs to be substantially improved in both theory and empirical study. The paper was judged quite incremental in comparison to the work of Öztoprak et al 2018 (where most of the theory was developed), while not showing improved empirical performance on the benchmarks.